# Study on the Impact of Climate Change on China’s Import Trade of Major Agricultural Products and Adaptation Strategies

**DOI:** 10.3390/ijerph192114374

**Published:** 2022-11-03

**Authors:** Chenchen Ding, Yong Xia, Yang Su, Feng Li, Changjiang Xiong, Jingwen Xu

**Affiliations:** 1College of Economics and Management, Xinjiang Agricultural University, Urumqi 830052, China; 2College of Business Administration, Xinjiang University of Finance & Economics, Urumqi 830012, China; 3Institute of Finance and Economics, Shanghai University of Finance and Economics, No. 777, Guoding Road, Yangpu District, Shanghai 200433, China

**Keywords:** climate change, agricultural products, import trade, adaptation strategies

## Abstract

With global warming, China’s agricultural products are facing severe production conditions and a complex international trade situation. In order to clarify the relationship between climate change and China’s agricultural trade, this paper uses the GTAP model to explore the impact of climate change on China’s agricultural trade from the perspectives of agricultural production and supply, energy substitution and trade policy. The results show that: (1) From the overall effect, the production supply risk and energy substitution risk caused by climate change have a positive impact on China’s import trade, among which the energy substitution risk has brought about an import trade growth of 38.050%, the production supply risk has brought about an import trade growth of 12.635%, and the trade policy risk has a negative impact, bringing about an import trade decline of 12.589%. (2) Under the impact of production and supply risks caused by climate change, the import volume of different industrial sectors has increased by varying degrees, including livestock products (16.521%) > food crops (14.162%) > cash crops (7.220%). The increase in import trade mainly comes from the United States (10.731%), Canada (10.650%) and Australia (9.455%). (3) Under the impact of energy substitution risk caused by climate change, the increase in import trade was concentrated in food crops (48.144%) and livestock products (42.834%), mainly from the United States (57.098%), the European Union (55.014%) and Canada (53.508%). (4) Under the impact of trade policy risks caused by climate change, the import trade of different industrial sectors showed a downward trend, with cash crops (13.039%) > livestock products (12.588%) > cash crops (12.140%). The countries and regions with significant decline in import trade were ASEAN (−46.131%) and the United States (−28.028%). The trade deficit shifted to surplus, and the terms of trade were improved. Therefore, this paper suggests that we should deal with the impact of climate change on agricultural trade by developing “climate smart” agriculture, actively responding to low-carbon trade measures, and establishing an agricultural trade promotion mechanism to address the risk of climate change.

## 1. Introduction

Since ancient times, the relationship between climate conditions and agricultural production has been closely linked, and its role is pivotal to the enhancement of agricultural productivity and is decisive to the increase or decrease in food production. In recent years, with global warming, glacial melting, sea level rise and extreme weather have seriously threatened the development of human society, and it not only destroys the natural ecological environment, but also aggravates the food crisis, which is a long-term problem that countries face and urgently need to solve [1]. Specifically, changes in climatic conditions such as temperature, precipitation, and radiation can change the growing conditions of crops and prolong or shorten the crop growth cycle [2]; for example, an increase in temperature can disrupt the crop growth cycle [3], a decrease in radiation hours can prevent grains from taking in sufficient energy, leading to a decrease in grain quality [4], and a lack or excess of water can inhibit normal crop growth and lead to a reduction in crop yield, etc. [5]. In the past 30 years, climate problems have led to a decline in grain production in some parts of China. For example, the increase in precipitation in southern China directly or indirectly led to the continuous decline of rice yield, and the decrease in precipitation in northwest China directly led to the decline in wheat yield [6]. On average, the annual grain output reduction caused by water-logging in China reached 6.502 million tons, accounting for 1.83% of China’s total annual grain output.

Agricultural trade is an important means to solve the problem of food security. Improving nutrition, promoting sustainable agriculture and eradicating world hunger while achieving food security are key to the United Nations Sustainable Development Goals (SDG) [7]. However, the problem is that the pattern of agricultural trade between countries is not stable. The impact of climate change on global agricultural output is becoming increasingly serious, which is bound to change the global trade structure and comparative advantage of agricultural products. On the one hand, climate change affects agricultural imports by affecting agricultural output. On the other hand, the increase in biomass energy in response to climate change will lead to increased demand and tight supply of grain and oil products. In addition, trade barriers such as “carbon tariffs” and “carbon standards” adopted by various countries in response to climate change have also brought more uncertainty to agricultural trade. Therefore, in order to deal with the impact of climate change through agricultural import trade, it is necessary to conduct in-depth discussion on the impact of climate change on agricultural import trade and clarify the impact scenario and extent. 

This paper intends to use the GTAP model to explore the impact of climate change on China’s agricultural trade from the perspectives of agricultural production, energy substitution and trade policy. It will not only find a way for China to mitigate the impact of climate change through agricultural product import trade, but also provide a reference for other countries in the world to address the impact of climate change.

## 2. Theoretical Foundation and Research Hypothesis

### 2.1. Theoretical Foundation

Agricultural production is naturally vulnerable and highly dependent on climatic conditions. Climate change is not only a unilateral change of temperature, water and other factors, but also indirectly leads to a certain change in the entire ecosystem, generating a series of chain reactions, such as a change in temperature will lead to the increase in pests and diseases, and the increase in fertilizer will lead to the decline of soil quality [8]. At the same time, climate change will also have a certain impact on the global trade pattern of agricultural products, which is mainly reflected in the impact of uncertain climate change on agricultural product output, leading to changes in the market price of agricultural products, even leading to changes in trade policies, creating artificial trade barriers, and ultimately affecting the import trade of agricultural products. 

According to existing research, the impact of climate change on agricultural trade mainly comes from two aspects: direct risk and indirect risk. First, from the perspective of direct risks, climate change induces the instability and imbalance of trade by impacting the supply and demand of agricultural products, thus affecting international trade of agricultural products. It is undeniable that high temperature has a positive impact on agricultural production to a certain extent, which may lead to the increase in agricultural output in some areas [9]. Taking China as an example, in recent decades, the rising temperature and radiation changes during the wheat growing period have increased the yield of wheat in northern China by 0.9–12.9% [10]. However, the negative impact cannot be ignored. The increase in average temperature will shorten the growth period of agricultural products and reduce the yield. Relevant studies also show that by the end of this century, climate change will lead to a reduction of rice yield by 2% to 16% and wheat yield by 3% to 19% in China [11]. The world will also lose 3 million tons of rice, 9 million tons of wheat and 2 million tons of soybeans annually due to climate change [12]. The decline in the output of agricultural products is bound to bring about the imbalance between supply and demand of agricultural products. Due to the impact of climate change, together with the increase in population and the reduction of planting area in various countries, many large agricultural export countries (such as the United States, Argentina, etc.) have reduced the supply of agricultural products, and the demand for agricultural products in various countries is growing day by day. While the imbalance between supply and demand leads to the tightening of global agricultural products supply, it also further increases the demand for agricultural products in China, which significantly increases the import of agricultural products in China. 

Second, from the perspective of indirect risks, climate change indirectly affects agricultural trade mainly through biomass energy substitution and low-carbon trade policies. On the one hand, with the formal entry into the force of the Kyoto Protocol in 2005, both developed and developing countries have actively taken various measures to deal with excessive CO_2_ emissions and other issues, launching a “blue sky defense war”, and developing biomass energy has become the consensus of all countries in the world. Fuel ethanol with corn, wheat, rice and other grain crops as the main raw materials is growing. With the fluctuation in oil price, fuel ethanol has become a direction of diversified energy alternative strategies in various countries by virtue of its clean, renewable and other characteristics [13]. The United States plans to replace 25% of crude oil imported from the Middle East with biomass fuel by 2025 and 30% of vehicle fuel with biomass fuel by 2030. China’s “Fourteenth Five Year Development Plan” plans that by 2025, the utilization of biomass fixed briquette fuel will reach more than 3 million tons, and the use of various biomass energies to replace standard coal will reach more than 4.8 million tons. Brazil and the European Union have formulated the “Sunshine Plan”, “Alcohol Energy Plan” and “Biofuel Strategy” to increase the application scale of biomass fuel. It is estimated that by 2035, biomass fuel will replace more than half of the world’s gasoline and diesel, with significant economic and environmental benefits [14,15]. 

The rapid development of bioenergy has led to increasing pressure on food supply. The data show that the prices of most staple agricultural products will rise in 2021, and the closing weekly average prices of soybean, wheat, corn and soybean oil futures contracts will be 11%, 34%, 40% and 37% higher than the same period last year, respectively. The international cotton index (SM level) was USD 2757 per ton, 46% higher than the same period last year. At the same time, the data show that the FAO food price index will increase by 27.2% year on year in 2021, reaching the highest level in a decade. The price of vegetable oil rose 65.8% year on year, a record high. The risks and uncertainties brought by the price of international bulk agricultural products on China’s future grain imports have indirectly pushed up the price of China’s agricultural products.

On the other hand, in response to climate change, the international community has taken joint action to control greenhouse gas emissions and has signed a number of multilateral environmental agreements. In this context, “low-carbon trade measures” to control greenhouse gas emissions have also been introduced. The United States has integrated climate change, trade and the environment into its new security concept, especially the “carbon tariff” policy, which will cause EU member countries and other developed countries to follow suit, which will undoubtedly bring huge risks to developing countries. According to the estimation of World Bank experts in 2009, if all countries levy a “border carbon tax” at the rate of 50 USD/ton of CO_2_, the average tariff faced by emerging developing countries such as Russia, China, South Africa and India will increase by 11.7%, 10.5%, 10.1% and 7.8%, respectively, which will undoubtedly increase the cost of agricultural product import trade and have a certain impact on agricultural products import. 

### 2.2. Research Hypothesis

#### 2.2.1. Climate Change-Production Supply-Import Trade

The output of crops depends not only on the improvement of modern agricultural technology, but also on the improvement of conditions [16]. Research shows that in the past 60 years, human activities have led to great changes in global climate conditions, the growth rate of agricultural productivity has slowed down, and the marginal role of science and technology in agricultural productivity has become smaller and smaller [17]. However, the negative effects of changes in natural conditions on agricultural production are increasing, mainly in the yield, quality and market of crops [18]. To be specific, since the 1960s, global warming has resulted in an average yield reduction of 5.3% in global corn, millet and rice. The yield reduction of corn is the largest, more than 5 percentage points, followed by wheat and finally rice, all of which have exceeded 4 percentage points [19]. Furthermore, more than half of the reduction in food production is due to the increase in ozone concentration [20,21], exacerbating the negative impact of climate change. In addition to the negative impact on main grain yield, climate warming also has a reduction effect on soybean yield, which has decreased by 4.5% in the past three decades [22]. From a regional perspective, climate change has led to an increase in extreme weather, diseases and pests. It has less impact on agriculture in developed regions at high latitudes, and more impact on developing countries at low latitudes. It has further widened the poverty gap between developed and developing countries, and has worsened the living conditions of people in underdeveloped countries [23]. Specifically, when the global temperature increases by about 1 °C, extreme weather will occur in West Africa in the Northern Hemisphere, resulting in more than a 5% reduction in sorghum and grain production [24]. At the same time, climate change causes poor effects of chemical fertilizers, pesticides and other chemicals, increases agricultural production costs [25,26], and has adverse effects on the ecological environment and soil quality [27]. Therefore, under the external environment of uncertain risks in the international society and economy, changes in temperature, precipitation and extreme weather brought about by climate change seriously affect food production, which in turn affects food output. Through the supply and demand theorem, it affects food prices and food demand, which in turn affects trade volume. 

Based on this, hypothesis H1 is proposed: climate change leads to production and supply risks, which have a positive impact on import trade volume,

#### 2.2.2. Climate Change-Energy Substitution-Import Trade

With global warming, excessive carbon dioxide emissions are generally considered to be the culprit leading to climate change, and the main source of carbon dioxide is fossil energy emissions. Reducing carbon dioxide emissions has become an important means and measure for countries to solve climate warming [28]. To this end, countries began to seek bio-energy to replace fossil energy in order to reduce carbon dioxide emissions. Among them, bio-energy uses corn, sugarcane, straw and other crops to process and synthesize ethanol as fuel. For example, in the United States, Europe and other countries, corn, wheat and sugarcane are mainly used as raw materials to synthesize bio-fuels. The rise of bio-energy has aggravated the already unbalanced relationship between food supply and demand, and the hidden danger of food security has also spread all over the world, especially in developing countries and underdeveloped regions. In order to achieve a low-carbon economy, developed countries use food crops as raw materials, indirectly reducing the supply of food crops and increasing the price of agricultural products on the market. The data show that more than 30% of grain will be used to produce bio-energy in the next decade, and the proportion of bio-ethanol production will further expand in the consumer market. Biomass energy has gradually replaced fossil energy as the main fuel for production and living, and has also become the trend of social and economic development, which is bound to further increase China’s demand for agricultural products. In the case of relatively reduced supply, and China’s basic principle of “not competing with people for food, not competing with grain for land” will further lead to the shortage of domestic agricultural products, an imbalance between supply and demand, and continuous price rise. It is bound to expand the import of agricultural products to meet people’s basic living needs and further provide the necessary basis for China’s energy transformation. 

Based on this, hypothesis H2 is proposed: climate change leads to the risk of energy substitution, which has a positive impact on import trade. 

#### 2.2.3. Climate Change-Trade Policy-Import Trade

In order to deal with global warming, countries have taken joint action and signed the United Nations Framework Convention on Climate Change, with a view to reduce global carbon dioxide emissions. At the same time, in terms of product import and export, each country has taken different measures to safeguard its own interests. Carbon tariff is an important tool for countries to implement low-carbon trade policies, that is, by identifying imported goods, to impose tariffs on products with high carbon emissions and no emission reduction measures [29]. China is the most populous developing country in the world. The imposition of carbon tariffs will have a serious negative impact on China’s import and export trade and economic development. Especially when carbon tariffs in the agricultural field are involved, they pose a huge challenge to China’s agricultural development. They will not only increase the cost of China’s agricultural production, weaken the comparative advantage of agricultural production, but also form a trade barrier to China’s agricultural imports and exports [30]. With the signing of cooperation agreements by various countries, non-tariff barriers have gradually replaced tariff barriers as new barriers hindering agricultural trade, which has a greater impact on the trade volume, welfare level and terms of trade of various countries, increases the unnecessary trade costs, and is not conducive to the sound development of the global economy [31]. 

Based on this, hypothesis H3 is proposed: climate change leads to trade policy risks, which have a negative impact on import trade volume.

Figure 1 shows the theoretical analysis model of climate change on China’s agricultural import trade.

## 3. Materials and Methods

### 3.1. Model Method

The GTAP model is a general equilibrium model widely used in academia, developed by Purdue University in the United States, and recognized by many scholars. This paper uses the GTAP model to analyze the impact of climate change on China’s agricultural imports. The GTAP model needs to meet the following five assumptions to more effectively simulate the trade situation of different regions and different sectors in the world [32]. First, the GTAP model needs to meet the perfectly competitive market state, that is, different countries and departments can participate in market trade at will. Second, the model should follow the condition that returns to scale remain unchanged, and the factor input and output show the same proportion of growth. Third, as a fixed factor of production, land should meet the basic conditions of non-mobility, while labor and capital can flow freely in different sectors, usually from low productivity sectors to high productivity sectors. Fourth, in the production process, the main goal is still to pursue profits, and different measures are taken to maximize profits and minimize costs. Fifth, the premise of a country’s import is that its domestic products are insufficient to meet domestic life and production, and there are no good substitutes in the country, so as to make up for the shortage of domestic products through trade imports between countries. 

In the GTAP model, the production function of the product is in the form of CES:(1)$$Q={\left(X^{\alpha -1/\alpha }+Y^{\alpha -1/\alpha }\right)}^{\alpha /\alpha -1}$$

In the formula, *X* is the combination of production factors, *Y* is the combination of intermediate products, and *α* indicates the substitution elasticity of production factors and intermediate products. The expression of combination *X* of production factors is:(2)$$X={\left(A_{1}^{\beta -1/\beta }+A_{2}^{\beta -1/\beta }+\cdots +A_{5}^{\beta -1/\beta }\right)}^{\beta /\beta -1}$$

In the formula, *A*_1_, *A*_2_, …, *A*_5_ represent production factors. β indicates the substitution elasticity between production factors.

GTAP model is a global model; thus, the price link equation is included in the trade module to describe the relationship between bilateral trade prices.
(3)$$PE_{i,r,s}^{fob}=PE_{i,r,s}\left(1+\tau_{i,r,s}^{e}\right)$$
(4)$$PM_{i,r,s}=PM_{i,r,s}^{cif}\left(1+\tau_{i,r,s}^{m}\right)$$
(5)$$PM_{i,r,s}^{cif}=PE_{i,r,s}^{fob}+t_{i,r,s}^{mg}{PWMG}_{i,r,s}$$

For goods in each sector *i*, $PE_{i,r,s}^{}$ represents the export price of exports from China to country s or region, $PE_{i,r,s}^{fob}$ represents the export FOB price, and $\tau_{i,r,s}^{e}$ represents export subsidies or taxes. Therefore, Formula (3) depicts the determination of export FOB price in export trade. $PM_{i,r,s}$ is the import price of the imported goods produced by China from S country or region, $PM_{i,r,s}^{cif}$ is the CIF import price, $\tau_{i,r,s}^{m}$ represents import duties or subsidies. Therefore, Formula (3) represents the determination of the price of imported goods in the domestic market of import trade. In the GTAP model, there is a difference between the FOB and CIF prices of goods. In the model, ${PWMG}_{i,r,s}$ is used to represent the national trade and transport prices between the two transaction nodes of region i and j, and the price change is one of the determinants of the difference between the CIF import price and the FOB export price, which is expressed by Formula (4). 

The GTAP model assumes that the economic system can finally reach a general equilibrium, and it analyzes the impact of different trade policies on import trade, based on the assumptions of perfectly competitive market, commodity market and factor market clearing, zero profit of producers, balance of income and expenditure of household sector, and free flow of labor and capital in various industries within the region.

In the process of solving the GTAP model, all functional forms are linearized to obtain the relationship between variable change rates. The terms of trade of China are defined as:(6) tor=psw−pdw

*psw* represents the change rate of the export price index of products manufactured in China, *pdw* represents the change rate of import price index of products manufactured in other countries imported by China, and *tor* represents the trade balance that is the difference between exports and imports.

In addition, the model uses the equivalent change method to measure the welfare change. The welfare impact caused by tariff change is represented by the change of distribution efficiency in the model:(7)$$EV_{s}\left(\tau_{i,r,s}^{m}\right)=\varphi_{s}\left(\tau_{i,r,s}^{m}PM_{i,r,s}^{cif}dQIM_{i,r,s}\right)$$

$\tau_{i,r,s}^{m}PM_{i,r,s}^{cif}$ represents the unit tariff income of commodity *i* imported from country s or region s to China when the ad valorem tariff level is $\tau_{i,r,s}^{m}$.

$dQIM_{i,r,s}$ represents the change level of the total import volume of commodity i from country or region s to China. $\varphi_{s}$ represents the scalar coefficient related to each exporting country; thus, the welfare change caused by the adjustment of the tariff rate in the model can be expressed by Formula (7).

### 3.2. Data Material Description

The GTAP database has been updated to the latest version of the 10th edition after continuous exploration by the academic community. Compared with the previous database, it has increased the number of countries and regions to 141, involving up to 65 industries, laying a more accurate and comprehensive research foundation for the development of international trade theory. The GTAPAgg covers the price, scale, output, import and export and other important contents of different industries and commodities in various countries. The GTAP database is mainly composed of input-output tables and trade data between countries or regions, which are used to analyze the economic development of countries and display specific information of different industries and trade between different countries. With its comprehensive database and industrial input and output, the GTAP model is often used to evaluate the impact of natural or manmade shocks such as policy change, climate change and economic change on different countries and sectors. The specific steps to build a GTAP model generally include the following two steps: one is to build a general equilibrium model that includes the consumption and expenditure of the government and consumers. Second, in order to more conveniently and accurately identify the degree of policy impact, a reasonable initial value should be selected for specific variables. When policies or the climate impact specific variables, the changes of specific variable values under the new equilibrium state are compared, and then the impacts of policy changes or climate changes on the import and export trade of different countries and different sectors are analyzed [33].

#### 3.2.1. Regional Division

For the purpose of analysis, the 141 countries are reclassified into nine regions in this paper, namely China, the United States, Russia, Canada, Australia, New Zealand, ASEAN, the EU, and other regions, as detailed in Table 1.

#### 3.2.2. Department Division

The details of different countries and different sectors in the GTAP database were collected in 2014. In order to portray and predict the impact of climate change shocks on agricultural trade in different economies in 2022, this paper uses the methods of Walmsley [34], Li Xinxing [35], etc., and the database of the French Center for International Economic Research to dynamically recurse variables such as GDP, population, capital, labor force, etc. to 2022. The GTAP model includes hundreds of major countries around the world, including 22 agricultural sectors such as wheat, rice, oil crops, other cereals, vegetables, fruits, and livestock products. This paper focuses on the impact of climate change on agricultural trade; thus, the industry sectors are divided into twelve industries in three sectors: food production, economic crop production, and meat and dairy production, as detailed in Table 2.

#### 3.2.3. Data Analysis

Figure 2 shows China’s imports of agricultural products in 2021. In 2021, China’s agricultural trade volume will be USD 304.17 billion, an increase of 23.2% year on year. Exports reached USD 84.35 billion, up 10.9%. Imports reached USD 219.82 billion, up 28.6%. The trade deficit was USD 135.47 billion, up 42.9%. From the perspective of the composition of imported varieties, cash crops amounted to USD 87.63 billion, accounting for 40%, livestock products amounted to USD 52.34 billion, accounting for 24%, food crops amounted to USD 20.07 billion, accounting for 7.9%, and other products amounted to USD 59.78 billion, accounting for 12%. Other products mainly include cotton yarn and other deeply processed products, which belong to industrial products in the import and export statistical classification and are not included in the scope of this study. (Data from Agricultural Trade Promotion Center of the Ministry of Agriculture and Rural Affairs of China.)

### 3.3. Simulation Scheme and Specific Scenario Design

#### 3.3.1. Production Supply Risk Simulation and Scenario Design

The changes in temperature, precipitation and extreme weather brought about by climate change will seriously affect food production and then affect food output. Through the supply and demand theorem, it will affect food prices and food demand and then affect trade volume. It is estimated that by the end of the 21st century, climate change will reduce the yield of major grains, with a median yield reduction of 1% to 3% every decade. The median yield reduction was 2.3% for corn, 3.3% for soybean, 1.3% for wheat and 0.7% for rice [36]. There was a significant negative correlation between temperature change and rice yield in southern China. The average temperature increased by 1 °C, and rice yield decreased by 2.61–3.57% [37]. From 1959 to 2007, the most serious impact of temperature change on rice yield was in northwest China, and the least was in southwest China. The order of temperature elasticity coefficient of rice yield was East China (−2.5%), North China (−2.3%), Northwest China (−3.2%), Southwest China (−1.6%), and Central South China (−2.6%) [38]. The impact of rising temperature on wheat yield in China is about 2.49% [39]. 

Therefore, in order to estimate the production and supply risks of agricultural products caused by climate change and the risk impact on China’s agricultural product import trade, this paper chooses “output” as the impact variable to investigate the changes in China’s agricultural product trade volume and trade pattern when the output of agricultural products decreases due to climate change. As shown in Table 3, Scenario 1, the trade changes of different agricultural sectors are simulated when the output decreases by 1%, 2% and 3%.

#### 3.3.2. Energy Substitution Risk Simulation and Scenario Design

The large-scale development of biomass energy has not only promoted the effective linkage between the energy market and the grain market, but it has also broken the supply and demand balance of the world agricultural product market for many years, further increasing the pressure on China’s agricultural product supply. The demand of fuel ethanol for corn and the demand for biodiesel for vegetable oil will keep the price of corn and oilseeds at a high level, reduce the planting of other crops, reduce the supply, increase the price, and cause a chain reaction. In 2022, the gap between China’s corn output and consumption will reach about 15 million tons, and the gap between soybean output and consumption will exceed 100 million tons. As the gap between supply and demand continues to expand, the domestic price of food crops has further increased.

Therefore, in order to estimate the energy substitution risk caused by climate change and the risk impact on China’s agricultural product import trade, this paper chooses “price” as the impact variable to investigate the changes in China’s agricultural product trade volume and trade pattern when climate change causes the agricultural product price to rise. As shown in Table 3, Scenario 2, the trade changes of different agricultural industry sectors are simulated when the price rises by 1%, 2% and 3%.

#### 3.3.3. Trade Policy Risk Simulation and Scenario Design

The rapid development of a low-carbon economy has impacted the international trade rules of agricultural products. Countries have formulated low-carbon trade policies, and non-tariff barriers have become an important tool to implement low-carbon trade policies. The research shows that non-tariff barriers gradually replace tariff barriers, which has a certain impact on agricultural trade [40]. Non-tariff barriers specifically refer to non-tariff measures, such as anti-dumping, countervailing, quotas, licenses, etc., which have the intention of trade protection, are beneficial to domestic manufacturers and discriminate against foreign manufacturers, thus significantly distorting and hindering international trade. Their functions are similar to tariffs [41]. That is, by increasing fixed costs and variable costs, the compliance costs borne by exporters to adapt to new standards are increased, which limits trade [42]. Sanitary and animal and plant quarantine are non-tariff barriers that are most used and have the greatest impact in the field of agricultural trade. In 2019, technical barriers to trade (TBT-SPS measures) accounted for 95.73% of non-tariff barriers, and the number of notifications reached 3209 [43,44,45]. Therefore, in order to estimate the trade policy risk caused by climate change and the risk impact on China’s agricultural product import trade, this paper chooses “non-tariff barriers” as the impact variable to investigate the changes in China’s agricultural product trade volume and trade pattern when climate change causes a rise in non-tariff barriers. As shown in Table 3, Scenario 3, the trade changes of different agricultural sectors are simulated when non-tariff barriers rise by 1%, 2% and 3%.

## 4. Multi-Scenario Simulation Results and Discussion

### 4.1. Scenario 1: Yield Change

Table 4 shows the impact of climate change on the import of different agricultural products due to the reduction of crop production in China. On the whole, when China’s crop production declined, the import volume of different industrial sectors showed an increase in varying degrees. In the grain production sector, rice has the largest change, with an average increase of 22.514% in rice imports. In the economic crop production sector, the change range of sugar is the largest, and the average increase in sugar imports is 13.869%. In the meat and milk production sector, the change range of meat products and other related products and dairy products is the largest. The average growth rate of the import of meat products is 24.559%, and the average growth rate of the import of dairy products is 20.166%.

Specifically, when China’s crop output decreased by 1%, the import growth of meat products and other related products, rice, and dairy products was the largest, with an increase of 12.280%, 11.257% and 10.083%, respectively. The growth of oil crops, cereals and other products and milk was relatively small, with an import growth of 0.318%, 1.586% and 1.957%, respectively. When China’s crop output decreased by 2%, the import growth of meat products and other related products, rice, and dairy products remained the largest, at 24.559%, 22.514% and 20.166%, respectively, while the import growth of oil plants was the smallest, at 0.635%. When China’s crop output decreased by 3%, the import growth of meat products and other related products, rice, and dairy products was 36.839%, 33.772% and 30.249%, respectively.

Table 5 shows that climate change has reduced China’s crop output, which in turn has an impact on the import volume of the main source countries of agricultural products. It can be found that China’s agricultural imports mainly come from the United States, Canada, Australia and other countries, with average growth rates of agricultural imports of 10.731%, 10.650% and 9.455%, respectively. The growth rates of agricultural imports from New Zealand and ASEAN are relatively small, with growth rates of 7.362% and 7.449%, respectively. Specifically, when the output of crops decreases by 1%, China’s imports of agricultural products increase. China’s imports of crops from the United States, Canada and Australia have the largest growth rates of 9.755%, 9.682% and 8.595%, respectively. When the output of crops decreased by 2%, China’s imports of crops from the United States, Canada and Australia increased the most, at 19.510%, 19.363% and 17.190%, respectively. When the output of crops decreased by 3%, the growth rates of the United States, Canada and Australia were 2.927%, 2.905% and 2.579%, respectively, which are smaller than when the output decreased by 1% and 2%. This was mainly due to the “chain effect” of climate change. On the one hand, global climate change has led to a decrease in the output of other countries, leading to a decrease in the output of major food-producing countries, thus reducing crop exports. On the other hand, other countries also need to import crops due to the reduction of grain production, which forms competition in the international market, and there is malicious monopoly trade behavior. In other words, China is limited by its land resource endowment, its demand for crops depends on imports to a certain extent, and its dependence on imports of agricultural products from the United States, Canada and Australia is high.

Table 6 shows the impact of climate change on the welfare and trade of countries due to the reduction of crop output in China. When the output decreases by 1%, China is in a trade deficit position in terms of trade balance. The import of agricultural products is greater than the export, and the trade balance is USD −45,456.57 million. The selected countries are in trade surplus. The trade surplus of the United States, the European Union and Canada was the largest, with USD 14,189.89 million, 9558.99 million and 1555.28 million, respectively. In terms of welfare level, the welfare level of China and Russia has improved, which are USD 22,653.69 million and 62.53 million, respectively. The welfare level of other countries has decreased. In terms of trade, China’s welfare conditions have improved to 0.28, and the welfare conditions of other countries have decreased.

When the output decreases by 2%, China is in a trade deficit in terms of trade balance, which is USD −90,913.14 million. The rest of the other countries are in trade surplus. The trade balances of the United States, the European Union and Canada rank first, which are USD 28,379.78 million, 19,117.98 million and 3110.56 million, respectively. In terms of welfare level, the welfare level of China and Russia has increased to USD 45,307.39 million and 125.06 million, respectively, while the welfare level of other countries has still decreased. China’s terms of trade have improved to 0.55, while the terms of trade of other remaining countries have decreased, with the United States having the largest reduction of 0.21.

When the output decreases by 3%, China’s trade deficit increases to USD −136,369.83 million in terms of trade balance, and the rest of the countries are in trade surplus. The trade surplus of the United States, the European Union and Canada was USD 42,569.70 million, 28,676.99 million and 4665.84 million. In terms of welfare level, the welfare water of China and Russia improved, which was USD 67,961.14 million and 187.59 million, respectively. The welfare level of other countries is still reduced. China’s terms of trade remained improved at 0.83, while other countries’ terms of trade decreased, and the United States’ terms of trade was the worst at −0.31.

### 4.2. Scenario 2: Price Change

Table 7 shows the impact of climate change on the import volume of different agricultural products due to the increase in crop prices in China. The results show that the rising price of agricultural products in China has prompted consumers to demand more foreign imports, which has led to an increase in import output of different industrial sectors. In the grain production sector, the import of rice and wheat increased significantly, with an average increase of 75.225% and 58.071%, respectively. In the economic crop production sector, the import of sugar crops, vegetables, fruits, nuts and other products increased significantly, with an average increase of 54.385% and 36.887%, respectively. In the meat and milk production sector, the meat products and other related products and dairy products changed significantly, with an average increase of 58.906% and 52.634% in their imports.

Specifically, when the price of China’s agricultural products rose by 1%, the import growth of rice, meat products and other related products, and wheat was the largest, at 37.613%, 29.453% and 29.035%, respectively, while the import growth of oil plants, milk, cereals and other related products was relatively small, at 4.332%, 4.310% and 5.569%, respectively. When the price of agricultural products in China rose by 2%, the import demand of rice, meat products and other related products, and wheat remained in the forefront, with import growth rates of 75.225%, 58.906% and 58.071%, respectively. The import growth of milk, oil plants, cereals and other related products was relatively small, with import growth rates of 8.619%, 8.664% and 11.137%, respectively. When the price of agricultural products in China rose by 3%, the import growth of rice, meat products and other related products, and wheat ranked in the top three, with import growth rates of 112.838%, 88.359% and 87.106%, respectively. The import growth of milk, oilseeds, cereals and other related products was 12.927%, 12.910% and 1.706%, respectively.

Table 8 shows the impact of climate change on the import volume of major source countries of agricultural products due to the rise in price of agricultural products in China. It can be found that the rise in price of China’s agricultural products has increased the demand for China’s agricultural products import. Among them, the import of agricultural products from the United States, the European Union and Canada has increased significantly, with an average growth rate of 57.098%, 55.014% and 53.508%, respectively. When the price of China’s agricultural products rose by 1%, China’s imports to the United States, the European Union and Canada increased by 28.549%, 27.507% and 26.754%, respectively. When the price of China’s agricultural products rose by 2%, China’s imports to the United States, the European Union and Canada increased by 57.098%, 55.014% and 53.508%, respectively. When the price of China’s agricultural products rose by 3%, China’s imports to the United States, the European Union and Canada increased by 85.647%, 82.521% and 80.262%, respectively. With the rise of prices, China’s imports of foreign agricultural products are also increasing, which shows that the prices of agricultural products play an important role in agricultural import and export trade. With the increase in price, the gap in the growth of China’s imports to various countries is also narrowed. This may be due to the intervention of national policies on the price of agricultural products, indicating that the stability of agricultural prices is an important means to optimize the trade structure.

Table 9 shows the impact of price changes on countries’ trade balance, welfare level and terms of trade. When the price of agricultural products in China rises by 1%, China and the United States are in trade deficit, USD −1531.83 million and −12,057.40 million, respectively. The rest of the countries are in a trade surplus. The trade balance of the EU, Canada and ASEAN is USD 3995.35 million, 2998.45 million and 2106.75 million, respectively. In terms of welfare level, the welfare level of China, Canada, ASEAN and the EU is negative, which is USD −9687.32 million, −1432.43 million, −2339.45 million and −11,895.81 million, respectively. The terms of trade of China, the United States and the EU have deteriorated, which are −0.27, −0.08 and −0.03, respectively.

When the price of China’s agricultural products rose by 2%, China’s trade deficit increased to USD −19,374.63 million in terms of trade balance, the United States changed from trade deficit to trade surplus, the trade balance was USD 17,403.41 million, and the EU changed from trade surplus to trade deficit, and the trade balance was USD −23,791.63 million. In terms of welfare level, China’s welfare level is still negative, which is USD −3063.66 million. The welfare level of the United States is reduced to USD −24,114.80 million. The welfare levels of Canada, ASEAN and EU changed from negative to positive, which are USD 5996.90 million, 4213.50 million and 7990.69 million, respectively. The terms of trade of China, the United States and the EU continued to deteriorate, which were −0.54, −0.16 and −0.05, respectively.

When the price rises by 3%, in terms of trade balance, the trade deficits of China, Canada, ASEAN and the EU increase to USD −29,061.96 million, −429.728 million, −7018.37 million and −35,687.54 million, respectively. The welfare level of China and the United States decreased to −4595.46 million and −36,172.24 million, respectively, and the welfare level of other countries increased. The terms of trade of China, the United States and the EU deteriorated, which are −0.80, −0.24 and −0.08, respectively.

### 4.3. Scenario 3: Economic Risk

Table 10 shows the impact of climate change on domestic agricultural products and then the adoption of non-tariff barriers and other measures to protect the development of domestic industries on the import volume of different industrial sectors. The results show that when the level of non-tariff barriers is raised, China’s rice imports are increasing, and the imports of other agricultural products are negatively related to the level of non-tariff barriers. Among them, when the non-tariff barrier rose, the average increase in rice import was 9.819%, and the import of wheat, beef, horse, mutton, grain and other related products decreased the most, with average decreases of −27.856%, −23.341% and −18.384%, respectively. The import of milk, vegetables, fruits, nuts, meat products and other related products declined relatively little, with an average decline of −3.147%, −7.384% and −9.009%, respectively. This shows that non-tariff barriers have a greater impact on rice and wheat, which will increase rice imports and reduce wheat imports. Rice and wheat are in a complementary relationship. China’s consumption of milk is relatively small compared with other agricultural products, and non-tariff barriers have a small decline in their imports.

Specifically, when the non-tariff barrier increases by 1%, the import of wheat, beef, mutton and horsemeat, cereals and other related products decreases by a relatively high rate of −13.928%, −11.670% and −9.192%, respectively, while that of milk, vegetables, fruits, nuts, meat products and other related products decreases by a relatively small rate of −1.573%, −3.692% and −4.505%, respectively. When the non-tariff barrier increases by 2%, the average decrease in the import of wheat, beef, mutton, horse meat, cereals and other related products is −27.856%, −23.341%, −18.384%, and the average decrease in milk, vegetables, fruits, nuts, meat products and other related products is −3.147%, −7.384%, −9.009%. When the non-tariff barrier is increased by 3%, the average decrease in the import of wheat, beef, mutton and horsemeat, cereals and other related products is −41.783%, −35.011% and −27.577%, and the average decrease in milk, vegetables, fruits, nuts, meat products and other related products is −4.720%, −11.075% and −13.514%.

Table 11 shows the impact of changes in non-tariff barriers on the import volume of major source countries of agricultural products. The results show that the changes of non-tariff barriers have a greater inhibitory effect on imports of ASEAN, the United States and other countries. The average decline of agricultural products in ASEAN, the United States and other countries was −46.131%, −28.028% and −31.226%, respectively. It greatly promoted China’s import of agricultural products to Australia and Russia, with an average increase of 8.819% and 7.546%. 

Specifically, when the non-tariff barrier increases by 1%, China’s imports of agricultural products to Australia and Russia will increase by 4.409% and 3.773%, respectively, and the imports of agricultural products to ASEAN and the United States will decrease by 23.065% and −14.014%, respectively. When non-tariff barriers increased by 2%, the growth of China’s agricultural imports to Australia and Russia increased by 8.819% and 7.546%, respectively, while ASEAN and the United States decreased by −46.131% and −28.028%, respectively. When the non-tariff barriers increased by 3%, the growth of China’s agricultural imports to Australia and Russia increased by 13.228% and 11.319%, respectively, and the decline of ASEAN and the United States was −69.196% and −42.042%, respectively. This shows that the United States and ASEAN, as the major countries of China’s agricultural imports, have raised non-tariff barriers to reduce the volume of agricultural imports from the United States and ASEAN, prompting China to increase imports of agricultural products in Australia and Russia, and reducing the risk of single import of agricultural products from the United States and ASEAN. At the same time, raising tariff barriers further reduces the dependence on foreign imports of agricultural products, increases the demand for domestic agricultural products, and is conducive to the development of domestic agricultural industry.

Table 12 shows the impact of changes in non-tariff barriers on the trade balance, welfare level and terms of trade of each country. When the non-tariff barrier rises by 1%, China, Canada, ASEAN and the EU are in a trade surplus with a trade balance of USD 3129.31 million, 1044.05 million, 2543.57 million and 13,596.04 million, respectively. The United States and Russia are in trade deficit with a trade balance of USD −11,453.08 million and −2358.43 million, respectively; In terms of welfare level, the welfare level of Russia, Canada and Australia has deteriorated, with welfare levels of USD −3015.76 million, −1747.15 million and −1631.03 million, respectively. The welfare levels of the United States, the European Union and China have improved, with welfare levels of USD 7954.11 million, 3169.77 million and 883.28 million, respectively. The terms of trade of Russia, Canada, Australia and other countries deteriorated to −0.61, −0.33, −0.56 and −0.07, respectively, and the terms of trade of China, the United States, New Zealand, ASEAN and the EU improved.

When non-tariff barriers rose by 2%, China, Canada, ASEAN and the EU had trade surpluses in terms of trade balance, which were USD 6258.62 million, 2088.10 million, 5087.13 million and 27,192.08 million, respectively. In terms of welfare level, the welfare level of Russia, Canada and Australia continued to deteriorate, which were, respectively USD −6031.52 million, −3494.29 million and −3262.06 million. The terms of trade of Russia, Canada and Australia deteriorated, which are −1.23, −0.65 and −1.11, respectively. When the non-tariff barrier rises by 3%, China, Canada, ASEAN and the EU are in a trade surplus, with a trade balance of USD 9387.96 million, 3132.15 million, 7630.6 million and 40,788.10 million, respectively, and the rest of the countries are in a trade deficit. In terms of welfare level, the welfare level of Russia, Canada and Australia deteriorated, respectively at USD −9047.29 million, −5241.44 million and −4893.09 million. The terms of trade of Russia, Canada and Australia are still deteriorating, respectively, at −1.84, −0.98 and −1.67.

The possible explanations for the above results are as follows:

Under the risk of production and supply, the growth rate of China’s agricultural product import trade is relatively small (12.635%). The reason may be that, on the one hand, climate change has reduced China’s crop output, prompted China to increase agricultural imports, and made China’s agricultural trade in the position of a trade deficit. China’s agricultural imports have improved China’s terms of trade and improved China’s welfare level to varying degrees. On the other hand, China’s land resources are limited, and it is possible to use limited resources to produce more agricultural products to ensure China’s food security by importing to meet domestic demand. By increasing the import of agricultural products and resource agricultural products, the trade structure of China’s agricultural products has been optimized, which plays an important role in the consumption of people’s daily life and easing the pressure on China’s resources. 

Under the risk of energy substitution, China’s import trade of agricultural products has a relatively large growth rate (38.050%). This shows that the price change of agricultural products caused by energy substitution is the key factor affecting import trade. The reason may be that rice and wheat, as the main food products in China, are also the raw materials of biomass energy. The domestic demand is large, and the impact of their price changes on import growth is particularly obvious. The production costs of rice and wheat in China are high, and they have no price advantage compared with the international market. When the price of agricultural products rises, consumers are more willing to choose imported products. In addition, from the perspective of changes in the trade pattern, when the climate change caused the price of China’s agricultural products to rise by 1 percentage point, the United States was in a trade deficit, but when the price rose by 2% and 3%, the United States changed from a trade deficit to a trade surplus. At the same time, the United States’ terms of trade did not improve. This is because the rise in the price of China’s agricultural products has increased the foreign demand for China’s agricultural products. At the same time, in order to reduce the import risk, we have chosen diversified import channels and methods for agricultural products import, not only limited to the United States, the European Union and other agricultural powers. In order to obtain the import orders of Chinese agricultural products and occupy the international agricultural market, the United States, the European Union and other countries have reduced prices to improve their agricultural competitive advantage and increase the export volume of agricultural products. As a result, although the United States and the European Union and other countries are in the situation of trade surplus, the terms of trade have deteriorated. 

Under the risk of trade policy, the overall import trade of China’s agricultural products showed negative changes. The reason may be that the promotion of non-tariff barriers is conducive to improving the welfare level and terms of trade of China’s agricultural products trade, reducing China’s agricultural imports and increasing exports. For different agricultural industry sectors, non-tariff barriers have a greater impact on wheat, beef, horse, mutton and other staple grain products and protein products. This is because wheat and rice are staple grains, and rice is planted more widely. Rice can be used as a complementary product of wheat to replace wheat. Therefore, the increase in non-tariff barriers will lead to an increase in rice imports and a decrease in wheat imports. At the same time, the most common measures in non-tariff barriers are sanitary and animal and plant quarantine measures; thus, they have a greater impact on the import of beef, horse and mutton.

## 5. Conclusions and Recommendations

### 5.1. Conclusion and Discussion

This paper focuses on the impact of climate change such as global warming, precipitation and extreme weather on China’s agricultural product imports by using the GTAP model. The research shows that climate change has led to changes in domestic agricultural product production and supply, price and energy substitution, and non-tariff barrier trade policies, thus affecting China’s agricultural product import trade. The conclusions are as follows,

First, research hypothesis 1 is valid, that is, climate change causes a decline in agricultural production, which has a relatively small positive impact on agricultural imports. When the output of agricultural products declined, China increased the import of agricultural products to make up for the gap in domestic demand. Among them, climate change has a relatively large impact on China’s food crops, meat products and dairy products, with a high growth rate in the import volume. Although China is in a trade deficit, due to the limited per capita land resources and the low level of agricultural scale in China, it is difficult to give consideration to food crops and resource agricultural products. The import of agricultural products makes full use of both domestic and foreign resources. The increase in agricultural product imports not only fills the gap caused by the decline in domestic output, but also it improves China’s terms of trade and domestic welfare.

Second, research hypothesis 2 is tenable, that is, climate change increases the demand for energy substitution, causing the rise in product prices, which has a relatively large positive impact on agricultural product import trade. Price, as the transmission medium of climate change, is a key factor affecting the import of agricultural products. Among them, rice and wheat have been greatly affected. The increase in prices has led to a large increase in their imports, indicating that food crops are necessities for people’s lives. Excessive price increases will weaken the comparative advantage of domestic food crops. At the same time, price increases are not conducive to the improvement of China’s welfare level and terms of trade. The rise in price of China’s agricultural products has aggravated the disadvantages of China’s agricultural products in the international market. 

Third, research hypothesis 3 is tenable, that is, climate change promotes the implementation of low-carbon trade policies, causes the rise in non-tariff barriers, and has a negative impact on agricultural imports. When non-tariff barriers to agricultural products rise, most of China’s agricultural imports are in a downward trend, indicating that non-tariff barriers have a certain protective effect on the development of domestic agriculture. The promotion of non-tariff barriers has guaranteed the sales of domestic agricultural products in the market, increased farmers’ income and production enthusiasm, and avoided the impact of foreign products on domestic products. At the same time, China’s trade balance has also changed from deficit to surplus, improving China’s terms of trade.

The discussion in this paper should be based on the predictable climate change scenario simulation. Global climate change may cause frequent extreme weather events, such as high temperature, drought and flood. The existing literature also finds that there is a significant correlation between extreme weather and food production. In addition, crude oil price, water resources and other factors have a profound impact on agricultural product import trade. Taking these factors into account in future research and incorporating them into the research model may lead to richer conclusions and provide more targeted and practical policy recommendations.

### 5.2. Policy Suggestions

Based on the previous research results, this paper proposes the following countermeasures and suggestions.

First, develop “climate smart” agriculture. “Climate smart” agriculture is a new agricultural development model proposed by humans to cope with climate change. FAO proposed in 2010 that developing countries need to develop “climate smart” agriculture to cope with the warming world and feed their growing population. China should continue to take “climate smart” development as the basis and improve the ability of agriculture to cope with climate change risks through improving carbon fixation agricultural varieties, strengthening “climate smart” technological innovation, and improving agricultural infrastructure.

Second, actively respond to low-carbon trade measures for agricultural products. As for “carbon tariff”, because it violates the basic rules of WTO and the principle of “common but differentiated responsibilities” of the Framework Convention, we should firmly oppose it and establish a domestic carbon tax mechanism as soon as possible. For the “carbon label” and “carbon certification” mechanisms, the use efficiency of fertilizers and pesticides should be improved by advocating green planting, properly purchase agricultural film and improve the treatment and utilization level of residual film, and establish and improve the low-carbon logistics system of agricultural products and other measures to eliminate the negative impact of low-carbon trade barriers on China’s agricultural products trade.

Third, we will establish a trade promotion mechanism for agricultural products to address climate change risks. On the one hand, it is necessary to establish a stable growth mechanism of agricultural product trade in response to climate change to ensure that the price of domestic grain and oil, vegetables, fruits and other important agricultural products will be stable under the influence of major adverse weather events. On the other hand, we should establish an early warning response mechanism for agricultural trade to deal with climate disasters and improve the rapid response and early warning capacity of agriculture to deal with climate change.

Climate change is not only an important problem for China, but also an important challenge for countries around the world to achieve sustainable development. China has always attached great importance to the impact, mitigation and adaptation of climate change, and is committed to promoting international and multilateral cooperation in global climate governance. At the same time, it also needs the joint efforts of all countries in the world to promote economic powers to move from confrontation to cooperation and to implement domestic agricultural adaptation measures to ultimately improve the adaptability and resilience of agriculture to climate change.

## Figures and Tables

**Figure 1 ijerph-19-14374-f001:**
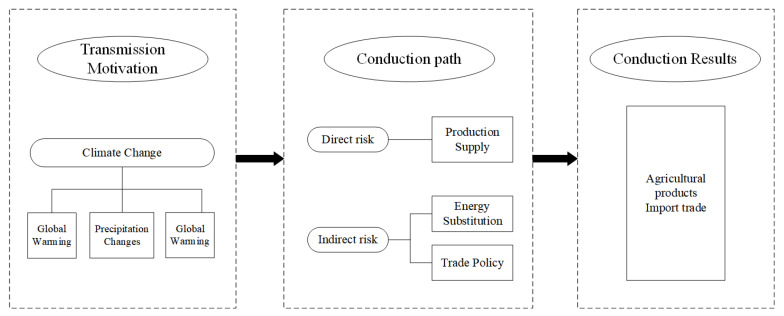
Theoretical analysis model of the impact of climate change on China’s agricultural import trade.

**Figure 2 ijerph-19-14374-f002:**
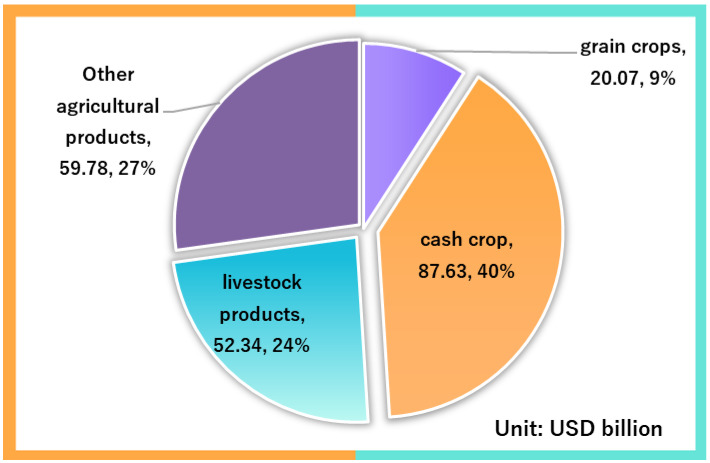
China’s agricultural product imports in 2021.

**Table 1 ijerph-19-14374-t001:** Country (region) division.

Regional Settings	Area Included
China	Mainland China
United States	United States
Russia	Russia
Canada	Canada
Australia	Australia
New Zealand	New Zealand
EU	France, Germany, Italy, Netherlands, Belgium, Luxembourg, Denmark, Ireland, Greece, Spain, Portugal, Austria, Sweden, Finland, Cyprus, Hungary, Czech Republic, Estonia, Latvia, Lithuania, Malta, Poland, Slovakia, Slovenia, Bulgaria, Romania, Croatia
ASEAN	Indonesia, Malaysia, Philippines, Thailand, Singapore, Brunei, Cambodia, Laos, Vietnam
Other Countries	Other countries and regions

Note: Data are aggregated by country or region in the GTAP database.

**Table 2 ijerph-19-14374-t002:** Agricultural industry sector division.

Industrial Sector	Industries Included
Food production sector	Rice, wheat, grains and other related products
Crop production sector	Oilseeds, sugar, vegetables, fruits, nuts, etc.
Meat and milk production sector	Beef, horse and sheep meat, milk, dairy products, meat products and other related products

**Table 3 ijerph-19-14374-t003:** Simulation scheme and scenario setting.

Simulation Scheme	Scenario Setting	Description of Simulation Results
Production supply	1. The yield of grain crops decreased by 1%; the yield of cash crops decreased by 1%; the yield of livestock products decreased by 1%. 2. The yield of grain crops decreased by 2%; the yield of cash crops decreased by 2%; the yield of livestock products decreased by 2%. 3. The yield of grain crops decreased by 3%; the yield of cash crops decreased by 3%; the yield of livestock products decreased by 3%.	1. For each specific scenario, simulate its independent impact on specific indicators. 2. The comprehensive impact of all specific scenarios of each simulation scheme is the total impact of the scheme. 3. The overall impact of the three schemes is the potential value of the project for setting indicators.

**Table 4 ijerph-19-14374-t004:** Effect of yield changes on imports of different agricultural products.

Industry Sector	1% Reduction in Production	2% Reduction in Production	3% Reduction in Production
Rice	11.257	22.514	33.772
Wheat	8.401	16.801	25.202
Grain and other related products	1.586	3.172	4.758
Oilseeds	0.318	0.635	0.953
Sugar	6.934	13.869	20.803
Vegetables, fruits, nuts, etc.	3.578	7.157	10.735
Beef, horse and lamb	8.723	17.446	26.169
Meat products and other related products	12.280	24.559	36.839
Milk	1.957	3.914	5.871
Dairy products	10.083	20.166	30.249

Note: Unit: %.

**Table 5 ijerph-19-14374-t005:** Impact of yield changes on imports from major source countries of agricultural products.

Country	1% Reduction in Production	2% Reduction in Production	3% Reduction in Production
United States	9.755	19.510	2.927
Russia	7.700	15.401	2.310
Canada	9.682	19.363	2.905
Australia	8.595	17.190	2.579
New Zealand	6.693	13.385	2.008
ASEAN	6.772	13.544	2.032
European Union	7.930	15.860	2.379
Other Countries	7.990	15.980	2.397

Note: Unit: %.

**Table 6 ijerph-19-14374-t006:** Impact of changes in production on welfare and trade across countries.

	1% Reduction in Production			2% Reduction in Production			3% Reduction in Production		
Country	Trade Balance	Benefit Level	Terms of Trade	Trade Balance	Benefit Level	Terms of Trade	Trade Balance	Benefit Level	Terms of Trade
China	−45,456.57	22,653.69	0.28	−90,913.14	45,307.39	0.55	−136,369.83	67,961.14	0.83
United States	14,189.89	−2622.75	−0.10	28,379.78	−5245.49	−0.21	42,569.70	−7868.24	−0.31
Russia	724.41	62.53	−0.01	1448.83	125.06	−0.02	2173.24	187.59	−0.02
Canada	1555.28	−131.39	−0.02	3110.56	−262.77	−0.05	4665.84	−394.16	−0.07
Australia	906.33	−78.14	−0.05	1812.65	−156.27	−0.09	2718.98	−234.41	−0.14
New Zealand	119.42	−12.80	−0.03	238.85	−25.60	−0.07	358.27	−38.40	−0.10
ASEAN	1459.62	−349.89	−0.03	2919.25	−699.78	−0.06	4378.87	−1049.67	−0.09
European Union	9558.99	−1226.37	−0.02	19,117.98	−2452.74	−0.05	28,676.99	−3679.12	−0.07

**Table 7 ijerph-19-14374-t007:** Impact of price changes on the import volume of different agricultural products.

Industry Sector	Price Increase of 1%	Price Increase of 2%	Price Increase of 3%
Rice	37.613	75.225	112.838
Wheat	29.035	58.071	87.106
Grain and other related products	5.569	11.137	16.706
Oilseeds	4.332	8.664	12.996
Sugar	18.128	36.257	54.385
Vegetables, fruits, nuts, etc.	12.296	24.591	36.887
Beef, horse and lamb	25.588	51.177	76.765
Meat products and other related products	29.453	58.906	88.359
Milk	4.310	8.619	12.929
Dairy products	26.317	52.634	78.951

Note: Unit: %.

**Table 8 ijerph-19-14374-t008:** Impact of price changes on the import volume of the main source countries of agricultural products.

Country	Price Increase of 1%	Price Increase of 2%	Price Increase of 3%
United States	28.549	57.098	85.647
Russia	23.812	47.624	71.436
Canada	26.754	53.508	80.262
Australia	19.205	38.409	57.614
New Zealand	19.845	39.690	59.535
ASEAN	22.176	44.351	66.527
European Union	27.507	55.014	82.521
Other Countries	24.793	49.587	74.380

Note: Unit: %.

**Table 9 ijerph-19-14374-t009:** Impact of price changes on welfare and trade across countries.

	Price Increase of 1%			Price Increase of 2%			Price Increase of 3%		
Country	Trade Balance	Benefit Level	Terms of Trade	Trade Balance	Benefit Level	Terms of Trade	Trade Balance	Benefit Level	Terms of Trade
China	−1531.83	−9687.32	−0.27	−19,374.63	−3063.66	−0.54	−29,061.96	−4595.46	−0.80
United States	−12,057.40	8701.70	−0.08	17,403.41	−24,114.80	−0.16	26,105.16	−36,172.24	−0.24
Russia	1547.78	2276.61	0.18	4553.22	3095.55	0.36	6829.83	4643.33	0.55
Canada	2998.45	−1432.43	0.10	−2864.86	5996.90	0.19	−4297.28	8995.35	0.29
Australia	953.28	1212.70	0.28	2425.41	1906.56	0.55	3638.12	2859.84	0.83
New Zealand	134.54	127.02	0.09	254.05	269.08	0.18	381.07	403.62	0.27
ASEAN	2106.75	−2339.45	0.07	−4678.90	4213.50	0.15	−7018.37	6320.26	0.22
European Union	3995.35	−11,895.81	−0.03	−23,791.63	7990.69	−0.05	−35,687.54	11,986.11	−0.08
Other Countries	1853.07	13,036.97	0.09	26,073.93	3706.15	0.17	39,110.91	5559.20	0.26

**Table 10 ijerph-19-14374-t010:** Impact of changes in non-tariff barriers on the import volume of different agricultural products.

Industry Sector	Non-Tariff Barriers Up 1%	Non-Tariff Barriers Up 2%	Non-Tariff Barriers Up 3%
Rice	4.909	9.819	14.728
Wheat	−13.928	−27.856	−41.783
Grain and other related products	−9.192	−18.384	−27.577
Oilseeds	−7.458	−14.916	−22.374
Sugar	−8.408	−16.817	−25.225
Vegetables, fruits, nuts, etc.	−3.692	−7.384	−11.075
Beef, horse and lamb	−11.670	−23.341	−35.011
Meat products and other related products	−4.505	−9.009	−13.514
Milk	−1.573	−3.147	−4.720
Dairy products	−7.427	−14.854	−22.281

Note: Unit: %.

**Table 11 ijerph-19-14374-t011:** Impact of changes in non-tariff barriers on imports from major source countries of agricultural products.

Country	Non-Tariff Barriers Up 1%	Non-Tariff Barriers Up 2%	Non-Tariff Barriers Up 3%
United States	−14.014	−28.028	−42.042
Russia	3.773	7.546	11.319
Canada	−5.180	−10.359	−15.539
Australia	4.409	8.819	13.228
New Zealand	−6.884	−13.768	−20.652
ASEAN	−23.065	−46.131	−69.196
European Union	−6.371	−12.742	−19.113
Other Countries	−15.613	−31.226	−46.839

Note: Unit: %.

**Table 12 ijerph-19-14374-t012:** Impact of changes in NTBs on countries’ welfare and trade.

	Non-Tariff Barriers Up 1%			Non-Tariff Barriers Up 2%			Non-Tariff Barriers Up 3%		
Country	Trade Balance	Benefit Level	Terms of Trade	Trade Balance	Benefit Level	Terms of Trade	Trade Balance	Benefit Level	Terms of Trade
China	3129.31	883.28	0.08	6258.62	1766.56	0.17	9387.96	2649.84	0.25
United States	−11,453.08	7954.11	0.30	−22,906.16	15,908.22	0.59	−34,359.26	23,862.34	0.89
Russia	−2358.43	−3015.76	−0.61	−4716.86	−6031.52	−1.23	−7075.29	−9047.29	−1.84
Canada	1044.05	−1747.15	−0.33	2088.10	−3494.29	−0.65	3132.15	−5241.44	−0.98
Australia	−205.01	−1631.03	−0.56	−410.03	−3262.06	−1.11	−615.05	−4893.09	−1.67
New Zealand	−25.40	24.10	0.06	−50.80	48.20	0.12	−76.19	72.30	0.17
ASEAN	2543.57	173.92	0.02	5087.13	347.83	0.04	7630.68	521.75	0.06
European Union	13,596.04	3169.77	0.06	27,192.08	6339.54	0.11	40,788.10	9509.34	0.17
Other Countries	−6271.03	−5811.27	−0.07	−12,542.05	−11,622.53	−0.14	−18,813.06	−17,433.79	−0.21

## Data Availability

Not applicable.

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
