# Peer review of "Study on the Impact of Climate Change on China’s Import Trade of Major Agricultural Products and Adaptation Strategies"

_ijerph, 2022, doi:10.3390/ijerph192114374_

Round 1

Reviewer 1 Report

Study on the Impact of Climate Change on China's Import
Trade of Major Agricultural Products and Adaptation Strategies

Comments

1.      The subject paper is a nice effort to reduce the economic risk by adopting measures to alleviate china's import of major agro products. The message and idea of the paper are appealing. The language of the paper needs some sort of improvement, however.

2.      Abstract is not self-explanatory. Data about results and policy implications are not clear in the abstract.

3.      Introduction: It is well-presented and coherently developed if some typos or grammatical mistakes are handled with just a thorough read by the authors.

4.      The authors properly address the main issues identified in the first review process. Nevertheless, it is not clear why he just calculates economic risk and not other risks.

5.      Material and methods:

6.      Material and method procedure is not clear. Are the data representative of china?

7.      Table 3 shows some Chinese words, that should be correct.

8.      Discussions: the authors fail to discuss the main important findings of the study and the implications of these findings for the policymakers.

9.      The section on implications for climate change adaptation needs further enrichment with some regional studies

10.  In Section Conclusion – the discussion about the study’s implications is not satisfactory. More discussion on why and how the results from this study would have implications for other countries is needed. Specifically, what lessons/strategies other countries can learn from this study?

Reviewer 2 Report

This paper identifies the impact of Climate change on imports, however, didn't establish a clear connection between this two. 

The theoretical pathway is not clear, that how this transformation will take place on import, there could be other factors too, especially the availability of products, international prices, relative prices, etc. 

How do the scenarios are compared and make a decision that is not clear from the paper? Please clearly mention this. 

The results are all not coming from the scenarios, the result should come from the analyzed data. 

There is some citation problem- following two types of citation methods at a time. Please avoid this. 

The citation is missing in line 62

Citation 10 to 15, use a single format 

Reviewer 3 Report

Dear authors,

The topic of the paper is interesting.

The main suggestions for changes are as follows:

Please highlight the objective of the paper at the end of the introduction section.

Please insert the research questions and the hypothesis/hypotheses of the research that will be later confirmed or denied.

Please translate into English the entire first row of Table 3.

I recommend a description or presentation of the methodology so as to highlight how the results in the main tables of the article were arrived at.

I recommend that for table 4 there are only 2 columns, the Industry Sector and the 1% reduction in production, the others being multiples of the first indicator, similarly for table 5.

Please add the limits of the research, either at the end of the conclusions or in the dedicated section.

Reviewer 4 Report

There are some Chinese characters in Table 3. Please write it in English 

Round 2

Reviewer 3 Report

Dear authors,

I appreciate the effort you put into making the suggested changes.

I consider that the paper meets the requirements for publication.

Best regards!